# Discovery of Pharmaceutical Composition for Prevention and Treatment in Patient-Derived Metastatic Medullary Thyroid Carcinoma Model

**DOI:** 10.3390/biomedicines10081901

**Published:** 2022-08-05

**Authors:** Hyeok-Jun Yun, Jin-Hong Lim, Sang-Yong Kim, Seok-Mo Kim, Ki-Cheong Park

**Affiliations:** 1Department of Surgery, Gangnam Severance Hospital, Yonsei University College of Medicine, Seoul 135720, Korea; 2Department of Surgery, Yonsei University College of Medicine, Seoul 120752, Korea

**Keywords:** patient-derived medullary thyroid carcinoma, cisplatin, sorafenib, cytochrome *c*, apoptosis

## Abstract

Medullary thyroid carcinoma (MTC) is a well-known neuroendocrine carcinoma, derived from C cells of the thyroid gland. Additionally, MTC is an uncommon aggressive carcinoma that metastasizes to lymph nodes, bones, lungs and liver. For MTC, the 10-year general survival ratio of patients with localized disease is about 95%, whereas that of patients with local phase disorder is around 75%. Only 20% of patients with distant metastasis to lung at diagnosis survive 10 years, which is notably lower than survival for well-differentiated thyroid carcinoma (WDTC). The management of MTC with distant metastasis to lung could be re-surgery or chemotherapy. In this research, we planned to assess the in vitro and in vivo combinational anticancer effect of a novel combination of low-dose cisplatin and sorafenib in patient-derived MTC. The patient-derived MTC cell lines YUMC-M1, M2, and M3 were isolated and treated with a combination of cisplatin and sorafenib or either agent alone. Cisplatin and sorafenib acted in combination to forward tumor restraint compared with each agent administered alone at a low dose. Therefore, a combination of cisplatin and sorafenib could be a new therapeutic approach for MTC.

## 1. Introduction

Thyroid carcinoma (TC), which accounts for more than 90% of total endocrine carcinomas, is the most common endocrine malignancy, and its incidence has increased over the last four decades [1]. TC is categorized into four main subtypes: papillary TC (PTC), follicular TC (FTC), medullary TC (MTC), and anaplastic TC (ATC) [2,3,4]. MTC progresses from C cells in the thyroid gland and is more aggressive and less differentiated than PTC or FTC [5]. MTC occurring from parafollicular cells is less typical than WDTC, making up between 2 [6] and 5% [7] of all thyroid malignancies. Among MTC patients, 13–15% experience distant metastasis, and the 10-year survival ratio was generally 20% [8,9]. Contrasted with differentiated thyroid carcinoma (DTC), where the iodine avidity of follicular cells makes even metastatic DTC compliant to therapy, parafollicular cells do not converge to iodine. Control options for recurrent or metastatic cases were re-surgery, radiotherapy and chemotherapy [10]. In addition, MTC is less common but encompasses aggressive malignancies with a short-term survival and frequent drug-resistance against anticancer drugs [11]. The fibroblast growth factor receptor (FGFR) signaling pathway is substantially induced in MTC, with the transforming growth factor β-mediated epithelial–mesenchymal transition (EMT) contributing to poor prognosis [12,13,14]. MTC usually has a good prognosis and positive response to therapy, whereas the presence of several metastatic MTC indicates a poor prognosis through early metastasis and recurrence, and these results can lead to patient death. The mission to improve the survival ratio and quality of life of patients with untreatable metastatic MTC has produced noteworthy studies on other therapies. The progress of targeted therapy has brought about major benefits in the control of such patients. Here, two drugs, cisplatin and sorafenib, have been approved for use in advanced or metastatic MTC [15,16]. According to the report of the NCI (National Cancer Institute), sorafenib is a multi-kinase inhibitor available for promoting apoptosis and suppressing tumor cell proliferation. This is a therapy for a certain type of thyroid cancer that targets inhibition of cancer cell growth and new blood vessels [17]. Furthermore, cisplatin is used to treat several types of cancer and is usually involved as a cytotoxic drug, increasing cancer cell death through injuring DNA and preventing DNA synthesis [18]. Although, the advance of cisplatin- and sorafenib-resistance in metastatic MTC is a crucial factor behind the breakdown of conventional cancer treatment.

In this study, we aimed to investigate whether aggressive MTC can be suppressed via the induction of cytochrome *c*-mediated apoptosis in response to the combinational anticancer treatment of cisplatin and sorafenib at a low dose compared to each agent administered alone. These findings could be clinically noticeable for the growth of novel combinatorial strategies effectively targeting highly malignant cells such as metastatic MTC. In particular, based on these findings, we are investigating the fundamental mechanism of anticancer drug resistance to forward the expansion of new pharmaceutical composition to address this problem.

## 2. Materials and Methods

### 2.1. Study Design and Ethical Considerations

This study was a retrospective, single-center analysis of patients diagnosed with MTC (between January 2003 and December 2019), as detailed in our previous study [19]. All procedures involving patients were performed in accordance with the institutional ethical standards, all applicable local/national regulations, and the guidelines of the 1964 Helsinki Declaration and its later amendments. In accordance with the Bioethics and Safety Act of Korea, formal written consent was not required for this type of retrospective, observational analysis. The study protocol was approved by the Institutional Review Board (IRB) of Severance Hospital, Yonsei University College of Medicine (IRB protocol: 3-2019-0281). Cell samples were obtained from patients at the Severance Hospital, Yonsei University College of Medicine, Seoul, Korea.

### 2.2. Patients

Patients who were diagnosed with MTC were pathologically confirmed either through surgery or open/core needle biopsy. Patients were followed up for at least 1–5 years or until death.

### 2.3. Patient Tissue Specimens and Clinical Course

Fresh tumor tissues were collected from patients with histologically and biochemical proven MTC who were treated at the Severance Hospital, Yonsei University College of Medicine, Seoul, Korea. Fresh tumors were collected throughout surgical excision of MTC metastatic and primary sites.

YUMC-M1 was a 24-year-old woman with sporadic MTC (stage II). The patient underwent bilateral total thyroidectomy with central compartment neck dissection and bilateral modified radical neck dissection and has been followed up for 71 months without recurrence or metastasis.

YUMC-M2 was a 19-year-old female MEN2B MTC (stage IVa) patient with RET-gene mutation. The patient underwent bilateral total thyroidectomy with bilateral modified radical neck dissection and later confirmed her pheochromocytoma and underwent left adrenalectomy. The patient has no metastasis to other organs, but she has been followed for 76 months without her serum calcitonin normalization.

YUMC-M3 was a 57-year-old man with sporadic MTC with lung, mediastinal and axillary lymph nodes metastases (stage IVc). The patient underwent bilateral total thyroidectomy with bilateral modified radical neck dissection, mediastinal dissection via partial sternotomy and axillary lymph node dissection. Since then, metastasis to the axillary lymph nodes, retropharyngeal space, and lung has progressed, and he was treated with a tyrosine kinase inhibitor, vandetanib.

### 2.4. Tumor Cell Isolation and Primary Culture

The patient-derived cancer cells were obtained from fresh tumors of patients. YUMC-M1, M2 and -M3 and -M1 were obtained from medullary thyroid cancer patients treated at the Severance Hospital, Yonsei University College of Medicine, Seoul, Korea. These patients were diagnosed with medullary thyroid cancer through fine-needle biopsy before surgery and histologically confirmed after surgery. Specimens were maintained and transferred to the laboratory. Fat and non-tumor parts were eliminated and washed using 1× Hank’s Balanced Salt Solution. Cell viability was evaluated with the trypan blue dye. Isolated tumor cells were authenticated with short tandem repeat profiling, karyotyping, and isoenzyme analysis. Mycoplasmal contamination was checked for with the Lookout Mycoplasma PCR detection kit (Sigma-Aldrich, St. Louis, MO, USA; MP0035). Further protocol details are as described in our previous article [19].

### 2.5. mRNA-Seq Data

We pre-processed the raw values from the sequencer to discard low attribute and adapter sequences before analysis and revised the procedure reads to Homo sapiens (GRCh37) using HISAT v2.1.0. The detailed protocol can be found in our previous article [19].

### 2.6. Statistical Analysis of Gene Expression Level

The comparative abundances of genes were calculated as read counts through StringTie. We carried out the statistical analysis to discover differentially expressed genes with evaluations of expression for each gene in the samples. The detailed protocol can be found in our previous article [19].

### 2.7. Hierarchical Clustering

Hierarchical clustering was carried out with complete connect and Euclidean distance as a determination of resemblance to show the expression patterns of separately shown transcripts, which are gratified with |fold change| ≥ 2 and separate *t*-test raw *p* < 0.05. The whole data analysis was carried out with R 3.5.1 (Vienna, Austria) (www.r-project.org, accessed on 1 October 2021).

### 2.8. Protein–Protein Interaction

Protein–protein interaction (PPI) was performed through the String database (https://string-db.org/, accessed on 20 October 2021) and visualized with Cytoscape 3.8.2 (https://cytoscape.org/, accessed on 20 October 2021). PPI was classified by a combination score (combination scores ≥ 0.4 were associated with the threshold value) and connecting whole genes counts (>3 counts).

### 2.9. Cell Culture

YUMC-M1, M2, and M3, MTC cell lines were obtained from patient specimens. These cells were cultured in RPMI-1640 medium including 15% fetal bovine serum (FBS; authenticated by short tandem repeat profiling/karyotyping/isoenzyme analysis). The detailed protocol can be found in our previous article [19].

### 2.10. Cell Viability Assay

Cell viability was estimated by the MTT (3-(4,5-Dimethylthiazol-2-yl)-2,5-Diphenyl tetrazolium Bromide) assay. Tumor cells were cultured in 96-well plates at 7 × 10^3^ cells per well. These cells were cultured overnight to accomplish 70–80% confluency. The indicated drugs were added to achieve final concentrations of 0–40 μM. The detailed protocol can be found in our previous article [19].

### 2.11. Cell Cycle Analysis Using Flow Cytometry

Cells were treated with a combination of cisplatin and sorafenib or either agent alone in RPMI-1640 medium containing 15% FBS for 40 h. The cells were then harvested by trypsinization and fixed in 70% ethanol. Further protocol details can be found in our previous article [19].

### 2.12. Immunofluorescence Analysis and Confocal Imaging

The level of cytochrome *c* was evaluated by the immunofluorescence staining assay. The primary antibody for the assay was anti-cytochrome *c* (1:25; Abcam, Cambridge, UK) incubated in 3% bovine serum albumin in PBS. Nuclei were stained with Hoechst 33342 (Life Technologies, Grand Island, NY, USA). Images for immunofluorescence were acquired using a confocal microscope (*LSM Meta 700*; Zeiss, Oberkochen, Germany) and were analyzed with Zeiss LSM Image Browser, version 4.2.0121.

### 2.13. Cellular Fractionation

Cellular fractionation was performed with the NEPER Nuclear and Cytoplasmic Extraction kit (Thermo Scientific, 78833, Waltham, MA, USA) in conformity with the manufacturer’s instructions.

### 2.14. Immunoblot Analysis

Primary antibodies against CHOP (C/EBP homologous protein, marker of ER stress) purchased from Cell Signaling (Cell Signaling Technology, Danvers, MA, USA); Bcl-2, cytochrome *c*, and histone H2B purchased from Abcam (Abcam, Cambridge, UK); and caspase 3 and β-actin purchased from Santa Cruz (Santa Cruz Biotechnology, CA, USA) were maintained overnight at 4 °C. Blots were developed with ECL reagents (Pierce) and exposed using Kodak X-OMAT AR Film (Eastman Kodak, Rochester, NY, USA) for 3–5 min.

### 2.15. Human MTC Cell Xenograft

All experiments were approved by the Animal Experiment Committee of Yonsei University. YUMC-M1, M2, and M3 patient-derived MTC cells (5.4 × 10^6^ cells/mouse) were cultured in vitro and then injected subcutaneously into the upper left flank region of female NOD/Shi-scid, IL-2Rγ KOJic (NOG) mice. After 15 days, tumor-bearing mice were grouped randomly (*n* = 10 per group) and treated with 5 mg/kg cisplatin (p.o.) alone, 80 mg/kg sorafenib (p.o.) alone, and 2.5 mg/kg cisplatin and 40 mg/kg sorafenib in combination. Tumor size was measured every three days using calipers. The detailed protocol can be found in our previous article [19]. Whole experiments were confirmed by the Animal Experiment Committee of Yonsei University.

### 2.16. Statistical Analysis

For statistical analysis of patient information, categorical changes were indicated as frequency and proportion, while summary statistics (median, range) were used to report consecutive results. Survival curves were produced through the Kaplan–Meier method established on the log-rank test. The detailed protocol can be found in our previous article [19].

## 3. Results

### 3.1. Patient Disease Characteristics

From January 2003 to December 2019, 143 patients were operated on for medullary thyroid cancer at Severance Hospital, Yonsei University College of Medicine, Seoul, Korea. Patient demographics and disease characteristics are presented in Figure 1A. The mean age of the patients was 48.9 years, and 62.2% of patients were females. The primary tumor diameter was 1.6 ± 1.5 cm, and tumor multiplicity was found in 26.6% of total patients. Positive nodal status was observed in 43.3% of patients, and *RET* mutation was confirmed in 25% patients. Median preoperative serum calcitonin and carcinoembryonic antigen levels were 151.5 pg/dL (range: 1.5–8030.0 pg/dL) and 10.0 ng/mL (range: 0.8–642.7 ng/mL), respectively. Serum calcitonin normalization (upper normal limit of 10 pg/mL) was found in 67.1% patients with MTC. Persistent and recurrent lesions were identified in 49 patients; of these patients, 32 underwent surgery, 3 received external radiation therapy, and 7 received chemotherapy. Of the patients who received chemotherapy, four received vandetanib, one received sorafenib and adriamycin, one received cabozantinib, and one received epirubicin and carboplatin. The median overall survival for 143 patients with MTC was 72.4 months (range: 4.2–215.5 months; Figure 1A). The patient survival rate was 90.7% at 5 years and 79.7% at 10 years after detection of TC (Figure 1B).

### 3.2. Features of Patient-Derived MTC Cell Lines

Varied MTC cell lines were isolated from the patient specimens (Figure 2A); YUMC-M1, M2, and M3 (first, second, and third isolated patient-derived MTC) were isolated from patients with MTC treated at the Severance Hospital, Yonsei University College of Medicine, Seoul, Korea. YUMC-M1, M2, and M3 were more aggressive than FTC, and metastasis was confirmed in these patients (Figure 2A). mRNA sequencing based on transcriptome analysis to identify a series of diversely expressed genes revealed that YUMC-M1, M2, and M3 cells were associated with a significant growth of the FGFR signaling pathway and EMT markers (*ZEB (Zinc finger E-box-binding homeobox)*, *SNAIL (Zinc finger protein SNAI1)*, and *TWIST (twist family bHLH transcription factor)*) were compared with YUMC-F1 (first isolated patient-derived FTC), as shown in Figure 2B. In MTC, the most remarkably raised genes were *FGF (Fibroblast growth factor)*, *FGFR, ZEB*, *SNAIL*, and *TWIST (twist family bHLH transcription factor)* (Figure 2B). Kyoto Encyclopedia of Genes and Genomes pathway enrichment analysis demonstrated the FGFR signaling pathway and EMT interaction (Figure 2C).

Taken together, the findings regarding MTC can be of outstanding value to therapeutic trials and the administration of patients with aggressive TC.

### 3.3. Combination of Cisplatin and Sorafenib Was More Efficacious Than Either Agent Alone

For evaluation of the combinational anticancer efficacy of cisplatin and sorafenib on patient-derived MTC cells, we investigated the proliferation of YUMC-M1, M2, and M3 cells in either agent combined or alone. Cell viability was restrained more efficaciously with a combination of cisplatin and sorafenib than with either agent alone in a dose-dependent manner (Figure 3A). Notably, the co-treatment had a lower half-maximal inhibitory concentration (IC_50_) than that with cisplatin or sorafenib treatment alone in YUMC-M1, M2, and M3 cells (Figure 3B). These results showed that co-treatment may offer a new clinical approach for targeting aggressive MTC with low doses of existing anticancer drugs.

### 3.4. Cisplatin and Sorafenib Co-Treatment Increased Nuclear Translocation-Mediated Apoptosis in YUMC-M1, M2, and M3 Cells

Next, we investigated the mechanism of the combinational anticancer effects of the co-treatment with cisplatin and sorafenib in YUMC-M1, M2, and M3 cells. We analyzed the expression levels of endoplasmic reticulum (ER) stress markers (C/EBP homologous protein (CHOP)) and apoptotic signaling pathway markers (B-cell lymphoma 2 (Bcl-2) and cleaved caspase 3) in YUMC-M1, M2, and M3 cells using flow cytometry, immunoblotting (whole-cell lysate or cellular fractionation), and immunofluorescence analyses. The combination of cisplatin and sorafenib at low doses significantly induced the sub-G_0_/G_1_ population, resulting in the induction of ER stress-mediated apoptosis (Figure 4A–C, right panel) in YUMC-M1, M2, and M3 cells compared with either agent alone (Figure 4A–C, left panel) and a strong inhibition of YUMC-M1, M2, and M3 cells. Immunoblot analysis of the protein expression levels indicated that the co-treatment with cisplatin and sorafenib led to a marked induction of CHOP and cleaved caspase 3 levels, which are active apoptosis markers associated with ER stress, whereas compared with either agent alone, the co-treatment at low doses resulted in a decrease in Bcl-2 level, which is a positive regulator of anti-apoptosis. Immunofluorescence analysis showed that cytochrome *c* was translocated to the nucleus, suggesting that the co-treatment with cisplatin and sorafenib increased apoptosis via a cytochrome *c*-dependent pathway (Figure 4D). Immunoblot analysis after cellular fractionation confirmed that cytochrome *c* was translocated to the nucleus after the co-treatment or treatment with either agent alone (Figure 4E). To sum up, these results showed that apoptosis was most caused by co-treatment with cisplatin and sorafenib through caspase- and cytochrome *c*-dependent pathways in patient-derived MTC cell lines.

### 3.5. Cisplatin and Sorafenib Co-Treatment Remarkably Restrained Tumor Growth in Mouse Xenograft Model with Patient-Derived MTC Cell Lines

To assessment the combinational in vivo anticancer effectiveness of the co-treatment with cisplatin and sorafenib, we developed a mouse xenograft model with YUMC-M1, M2, and M3 patient-derived MTC cell lines (Figure 5). We discovered that cisplatin or sorafenib alone could restrain tumor growth, but compared with either agent alone, the co-treatment clearly increased tumor shrinkage at low doses (Figure 5A,D,G). Appreciably, YUMC-M3 was not markedly impacted by cisplatin or sorafenib alone. Moreover, mice in the cisplatin and sorafenib co-treatment group had significantly smaller excised tumor weight than that of mice administered with cisplatin or sorafenib alone (Figure 5B,E,H). There was no proof of systematic toxicity and death in any group. The body weight of mice did not remarkably differ among all groups (Figure 5C,F,I).

Taken together, the combination of cisplatin and sorafenib indicated potent anticancer effects in an aggressive patient-derived MTC cell xenograft model.

## 4. Discussion

The survival ratio of thyroid carcinoma is well-known as being the best among general endocrine-mediated malignancies [20], and its onset rate is consistently growing worldwide, as well as specifically in the Korea [20,21,22,23]. Thyroid carcinoma is poised to become the fourth leading carcinoma worldwide [22]. For the past 20 years, the global age-standardized incident ratio of thyroid carcinoma has raised by roughly 25% [22,24,25]. This worldwide expansion has been ascribed to, e.g., raised factors, early disclosure of cancer, induced environmental hazard elements, and iodine degrees [22]. MTC is known to be an uncommon endocrine carcinoma that occurs in thyroid parafollicular C cells [26,27]. In particular, various investigations have shown that advanced or metastatic MTC has demonstrated refractory to almost medical therapies [28,29]. Metastasis of the MTC was via blood and lymphatic vessels, which had more aggressive clinical behavior than DTC. Rearranged during Transfection (RET) gene mutation is known as a central molecular change in MTC [30,31,32]. In germline RET mutation, MTC is classified under three different phenotypes: familial MTC, and multiple endocrine neoplasia types 2A and 2B (MEN2) [33,34]. Moreover, somatic RET mutation stands as the major molecular change in almost all cases [35,36,37]. In surgical cases, patients could choose whole thyroidectomy with prophylactic or remedial pivotal section lymph node resection [38]. However, unfortunately, in the case of large metastasis, especially in those with remarkably induced tumor growth, extra remedies were required. Moreover, systemic treatment of advanced or metastatic MTC is limited at present to administration of a tyrosine kinase inhibitor (TKI) [39,40,41,42]. Here, we report the combinational anticancer efficacy of cisplatin and sorafenib at a low dose compared with each treatment agent alone in patient-derived advanced MTC. Cytochrome *c* release from mitochondria is a critical event for apoptotic death [43,44]. Distinctly, cytochrome *c* release was significantly induced in the caspase-dependent apoptosis signaling pathway [45,46]. We indicted that release of cytochrome *c* from mitochondria most significantly increased apoptosis, and progressively amassed in the nucleus as proven by both immunofluorescence and cellular fractionation after medication of cisplatin and sorafenib at a low dose. Clinically aggressive MTCs lead to poor prognosis [47], considered as refractory carcinoma subtypes owing to their aggressive behavior [48]. Accordingly, there is an imminent need to investigate new clinical approaches for the remedy of aggressive MTCs.

In this research, we showed a prospective new therapy scheme for advanced MTC (Figure 6). Of course, our proposal might be the tip of the iceberg to numerous clinical approaches for therapy of advanced MTC. In particular, the study of patient-derived MTC was difficult and limited. Nevertheless, if these challenges continue to pile up, there could be a breakthrough for a clinical solution of advanced MTC. Clinically, these findings convey critical implications for the progress of novel combinatorial strategies that target selective drug-resistant cancer cells.

## 5. Conclusions

According to the guidelines of the National Cancer Institute, cisplatin and sorafenib are already well-known anticancer drugs. Although, in conventional cancer therapy, there is no definite solution for treatment of recurrent or metastasis by refractory MTC. In these research findings, the combinational anticancer consequences of the cisplatin and sorafenib were more capable at low doses of each agent alone against patient-derived metastatic MTC. These results could be practical in the design of prospective therapeutic trials and aid the expansion of valuable therapies for patients with refractory MTC. However, research on patient-derived refractory cancer cells is challenging and limited. However, these challenges could be cause to a breakthrough in, and clinical solution to, refractory cancer therapy.

## Figures and Tables

**Figure 1 biomedicines-10-01901-f001:**
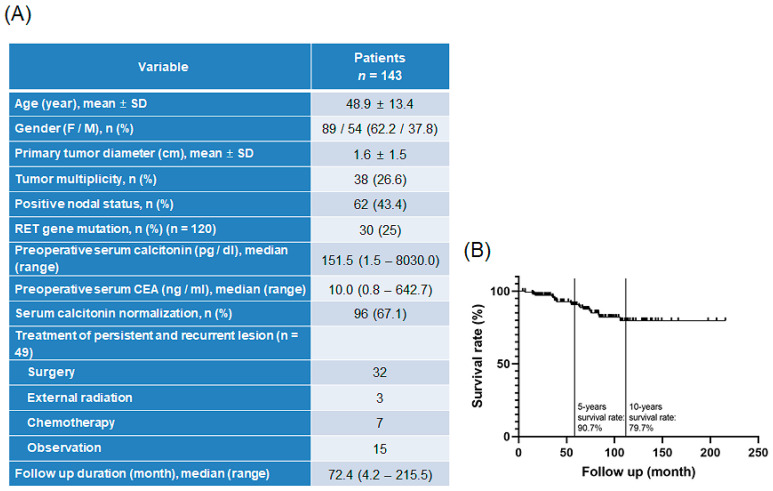
Information about patients treated for MTC in Severance Hospital: (**A**) patient characteristics and clinical features, and (**B**) overall survival rate of patients with MTC. MTC, medullary thyroid carcinoma.

**Figure 2 biomedicines-10-01901-f002:**
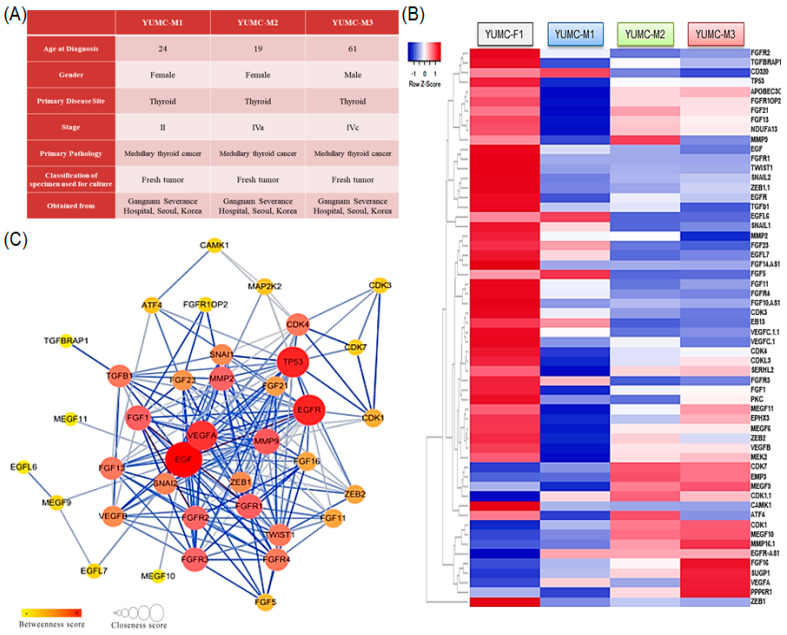
Characteristics of all examined MTC cell lines: (**A**) Characteristics of patient-derived subtypes of MTC cell lines. (**B**) Hierarchical clustering of annotated genes revealing distinct gene expression. Gene expression profiles of patient–derived MTC cells. (**C**) KEGG pathways connecting to the identified proteins reported in STRING database and connecting to MTC are highlighted (color coded included in the figure). Protein–protein interactions were regenerated based on both known and predicted interactions in MTC with STRING analysis. MTC, medullary thyroid carcinoma; KEGG, Kyoto Encyclopedia of Genes and Genomes.

**Figure 3 biomedicines-10-01901-f003:**
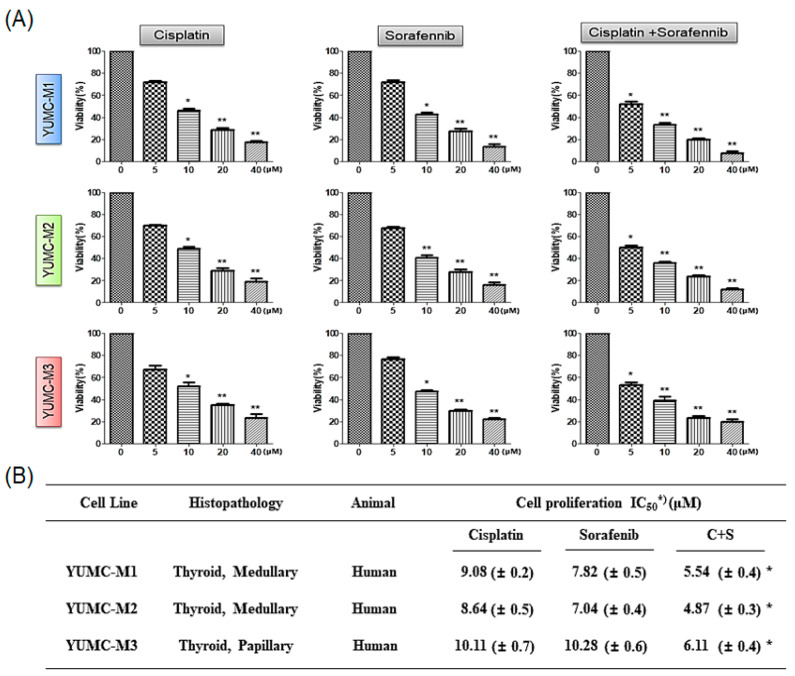
Combinational anticancer effects of cisplatin and sorafenib in patient-derived MTC YUMC-M1, M2, and M3 cells compared to the effects of each agent alone. (**A**) Cell viability with cisplatin and sorafenib combined or each agent alone. All experiments were repeated at least thrice. Data represent mean ± standard deviation. * *p* < 0.05 and ** *p* < 0.01 versus control. (**B**) IC_50_ values for the combination of cisplatin and sorafenib in patient-derived MTC, YUMC-M1, M2, and M3 cells. Each data point signifies the mean of three independent MTT assays, performed in triplicate. SEM, standard error of the mean; MTT, 3-(4,5-dimethylthiazol-2-yl)-2,5-diphenyltetrazolium bromide; IC_50_, half-maximal inhibitory concentration. The asterisk indicates lowest half-maximal inhibitory concentration.

**Figure 4 biomedicines-10-01901-f004:**
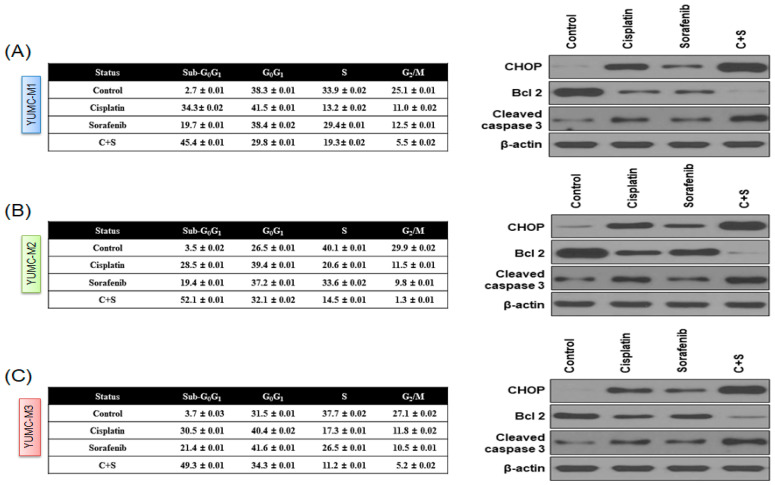
Apoptosis analysis by quantitation of DNA content with propidium iodide and immunoblot analysis. (**A**–**C**) Flow cytometry (left panel) and immunoblot analysis (right panel) of YUMC-M1, M2, and M3 cells. Cells were exposed to the indicated inhibitors, harvested, and stained with propidium iodide before analysis by flow cytometry and FlowJo version 8, Ashland, Dickinson and Company; 2021. Immunoblot analysis of the markers of endoplasmic reticulum stress, apoptosis, and anti-apoptosis in patient-derived MTC cell lines, YUMC-M1, M2, and M3. (**D**,**E**) Cisplatin and sorafenib highly induced nuclear translocation of cytochrome *c*-mediated apoptosis in patient-derived MTC cell lines, YUMC-M1, M2, and M3. (**D**) Immunofluorescence examined at 40× magnification; scale bar: 20 μm; and (**E**) subcellular fractionation analysis. Cytochrome *c* was most translocated and accumulated in the nucleus in the cisplatin and sorafenib co-treatment group.

**Figure 5 biomedicines-10-01901-f005:**
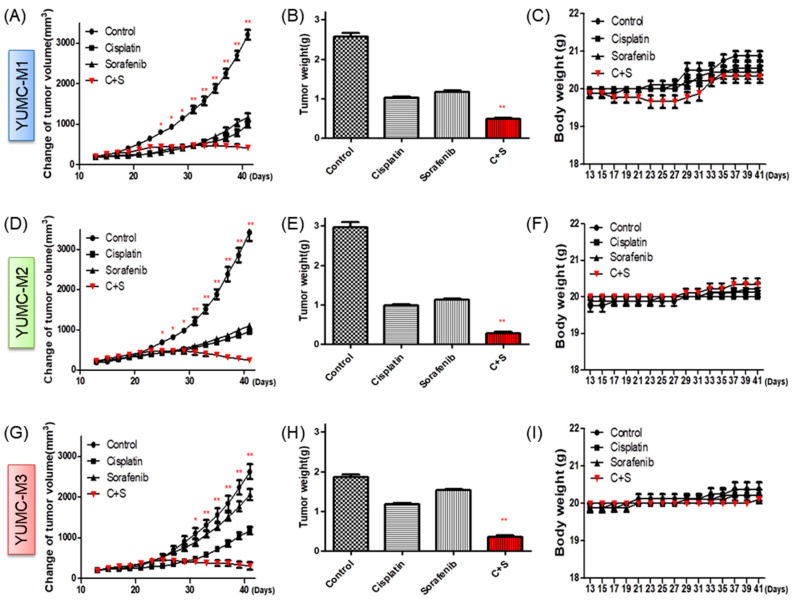
Anticancer effects of cisplatin and sorafenib in combination or with each agent alone in patient-derived MTC cell lines, YUMC-M1, M2, and M3 cell xenografts in vivo. (**A**,**D**,**G**) Change in tumor volume. (**B**,**E**,**H**) The dissected tumor weight. (**C**,**F**,**I**) Change in whole body weight. Cisplatin and sorafenib had no significant effect on mouse body weight. * *p* < 0.05 and ** *p* < 0.01, compared with control.

**Figure 6 biomedicines-10-01901-f006:**
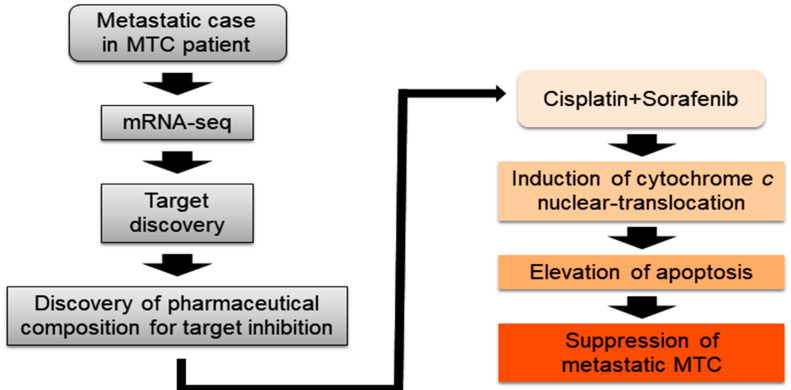
Scheme of study design.

## Data Availability

The data presented in this study are available on reasonable request from the corresponding author.

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
