# Peer review of "Discovery of Pharmaceutical Composition for Prevention and Treatment in Patient-Derived Metastatic Medullary Thyroid Carcinoma Model"

_biomedicines, 2022, doi:10.3390/biomedicines10081901_

Round 1

Reviewer 1 Report

The submitted manuscript entitled "Discovery of pharmaceutical composition for preventing and treating in patient-derived metastatic medullary thyroid carcinoma model" and presented by Yun et al., shows the role of Cisplatin and sorafenib to administered alone at low dose. The work presented here is fine but authors need to modify their manuscript in introduction and conclusion sections because both these two sections are very week. Authors need to take help with previous published literature and add the relevant information’s in this area of research. Below is the detail where authors need to do work on their manuscript. 

1. Introduction must need to add more relevant information for to show the problem and their solution with describing the novelty of the work. 

2. The resolution of Fig.1c is not clear and it must need to add more good resolution picture.  

3. The scale bars in Fig.1D is not clear and it must need to add visible scale bars for the readers. 

4. Discussion need to add the pictorial mechanism for more clear picture for the authors. 

5. In Conclusion authors need to add the finding of their experiments with conclusion also need to add the future prospective of their work. 

Author Response

Reviewer #1

The submitted manuscript entitled "Discovery of pharmaceutical composition for preventing and treating in patient-derived metastatic medullary thyroid carcinoma model" and presented by Yun et al., shows the role of Cisplatin and sorafenib to administered alone at low dose. The work presented here is fine but authors need to modify their manuscript in introduction and conclusion sections because both these two sections are very week. Authors need to take help with previous published literature and add the relevant information’s in this area of research. Below is the detail where authors need to do work on their manuscript. 

Reply: I don't know how to thank you enough for reviewing our manuscript. I agree with you completely and follow your professional opinion. I have made the suggested correction. Corrected sentence were indicated red color and the highlighted sentences with track-change. Thank you again for your review. I hope you are always healthy and happy!!

  1. Introduction must need to add more relevant information for to show the problem and their solution with describing the novelty of the work. 

Reply: Thank you for your comment. Follow expert your advice, I have added the suggested correction in section of introduction. Thank you again for your expert advice.

  1. The resolution of Fig.1c is not clear and it must need to add more good resolution picture.  

Reply: Thank you for your comment. Follow expert your advice, I have made the suggested correction in figure 1C. The resolution of ‘Figure 1C’ was increased as high as possible.  

  1. The scale bars in Fig.1D is not clear and it must need to add visible scale bars for the readers. 

Reply: Thank you for your comment. In my humble opinion, your commented ‘Fig.1D’ means maybe figure 4D. Follow expert your advice, I have made the suggested correction in figure 4D. Thank you again for your expert advice.

  1. Discussion need to add the pictorial mechanism for more clear picture for the authors. 

Reply: Thank you for your comment. Figure 6,’ Scheme of study design’ was added in section of ‘discussion’. Thank you again for your expert advice.

  1. In Conclusion authors need to add the finding of their experiments with conclusion also need to add the future prospective of their work. 

Reply: Thank you for your comment. Follow expert your advice, I have made the suggested correction in section of ‘conclusion’. Thank you again for your expert advice.

Reviewer 2 Report

The manuscript proposes the use of a combined treatment of cisplatin and sorafenib to improve the prognosis of patients with untreatable metastatic Medullary thyroid carcinoma (MTC).

The anticancer effects of the proposed combination (cisplatin and sorafenib) have been investigated using in vitro (cellular models derived prom patients affected with MTC) and in vivo assays (mice) assays. For comparison purposes, the effects of the using of a single agent (cisplatin or sorafenib) have been also considered by authors.

The manuscript is interesting and well organized. However some minor points need to be improved:

•             the quality of the figures appear with a low resolution, I suggest to improve it,

•             The introduction and conclusion sections need to be extended to well support the experimental design and the results obtained respectively.

Author Response

Reviewer #2

The manuscript proposes the use of a combined treatment of cisplatin and sorafenib to improve the prognosis of patients with untreatable metastatic Medullary thyroid carcinoma (MTC). The anticancer effects of the proposed combination (cisplatin and sorafenib) have been investigated using in vitro (cellular models derived prom patients affected with MTC) and in vivo assays (mice) assays. For comparison purposes, the effects of the using of a single agent (cisplatin or sorafenib) have been also considered by authors.

The manuscript is interesting and well organized. However some minor points need to be improved:

Reply: I don't know how to thank you enough for reviewing our manuscript. I agree with you completely and follow your professional opinion. I have made the suggested correction. Corrected sentence were indicated red color and the highlighted sentences with track-change. Thank you again for your review. I hope you are always healthy and happy!!

  1. The quality of the figures appear with a low resolution, I suggest to improve it,

Reply: Thank you for your comment. Follow expert your advice, I have corrected whole figure in this article. Thank you again for your expert advice.

  1. The introduction and conclusion sections need to be extended to well support the experimental design and the results obtained respectively.

Reply: Thank you for your comment. Follow expert your advice, I have made the suggested correction in section of ‘introduction and conclusion’. Thank you again for your expert advice.

Round 2

Reviewer 1 Report

The revised manuscript still need more elaboration in introduction and conclusion sections. Authors need to add more relevant information and future prospective of their work. After these elaboration this manuscript is publishable to this journal

Author Response

Reviewer #1, R2

The revised manuscript still need more elaboration in introduction and conclusion sections. Authors need to add more relevant information and future prospective of their work. After these elaboration this manuscript is publishable to this journal

Reply: I don't know how to thank you enough for reviewing our manuscript. And I'm sorry I didn't contribute to your expert advice. I did my best to correct in section of ‘introduction’ and‘conclusion’. Corrected sentence were indicated red color and the highlighted sentences with track-change. Thank you again for your review. I pray that you are always healthy and happy!!
